# Systematic Review of Lung Cancer Screening: Advancements and Strategies for Implementation

**DOI:** 10.3390/healthcare11142085

**Published:** 2023-07-21

**Authors:** Daniela Amicizia, Maria Francesca Piazza, Francesca Marchini, Matteo Astengo, Federico Grammatico, Alberto Battaglini, Irene Schenone, Camilla Sticchi, Rosa Lavieri, Bruno Di Silverio, Giovanni Battista Andreoli, Filippo Ansaldi

**Affiliations:** 1Regional Health Agency of Liguria (ALiSa), 16121 Genoa, Italy; daniela.amicizia@unige.it (D.A.); francesca.marchini@alisa.liguria.it (F.M.); matteo.astengo@alisa.liguria.it (M.A.); federico.grammatico@alisa.liguria.it (F.G.); alberto.battaglini@alisa.liguria.it (A.B.); irene.schenone@alisa.liguria.it (I.S.); camilla.sticchi@alisa.liguria.it (C.S.); rosa.lavieri@alisa.liguria.it (R.L.); bruno.disilverio@alisa.liguria.it (B.D.S.); giovannibattista.andreoli@alisa.liguria.it (G.B.A.); filippo.ansaldi@alisa.liguria.it (F.A.); 2Department of Health Sciences (DiSSal), University of Genoa, 16132 Genoa, Italy

**Keywords:** lung cancer, screening program, low-dose computed tomography (LDCT), cost-effectiveness, mortality reduction, healthcare

## Abstract

Lung cancer is the leading cause of cancer-related deaths in Europe, with low survival rates primarily due to late-stage diagnosis. Early detection can significantly improve survival rates, but lung cancer screening is not currently implemented in Italy. Many countries have implemented lung cancer screening programs for high-risk populations, with studies showing a reduction in mortality. This review aimed to identify key areas for establishing a lung cancer screening program in Italy. A literature search was conducted in October 2022, using the PubMed and Scopus databases. Items of interest included updated evidence, approaches used in other countries, enrollment and eligibility criteria, models, cost-effectiveness studies, and smoking cessation programs. A literature search yielded 61 scientific papers, highlighting the effectiveness of low-dose computed tomography (LDCT) screening in reducing mortality among high-risk populations. The National Lung Screening Trial (NLST) in the United States demonstrated a 20% reduction in lung cancer mortality with LDCT, and other trials confirmed its potential to reduce mortality by up to 39% and detect early-stage cancers. However, false-positive results and associated harm were concerns. Economic evaluations generally supported the cost-effectiveness of LDCT screening, especially when combined with smoking cessation interventions for individuals aged 55 to 75 with a significant smoking history. Implementing a screening program in Italy requires the careful consideration of optimal strategies, population selection, result management, and the integration of smoking cessation. Resource limitations and tailored interventions for subpopulations with low-risk perception and non-adherence rates should be addressed with multidisciplinary expertise.

## 1. Introduction

Lung cancer is a significant global health issue and is one of the leading causes of cancer-related deaths worldwide, for both men and women [1,2,3]. 

According to the World Health Organization (WHO), lung cancer accounts for 2.2 million new cases annually, representing 11.7% of all cancer cases, and 1.8 million deaths, representing approximately one-fifth of all cancer deaths, significantly more than those of breast and colorectal cancers combined [1,3,4].

Data reported by the American Cancer Society on 12 January 2023 provided estimates for the entire year of 2023. These estimates indicate that in the year 2023, there will be 238,340 new lung cancer cases in the United States, with 117,550 in men and 120,790 in women. Additionally, 127,070 deaths are attributed to lung cancer, with 67,160 in men and 59,910 in women [1].

The incidence of lung cancer varies widely worldwide, with the highest rates observed in Western countries such as the United States, Canada, and Europe and lower rates observed in developing countries [5].

Smoking is the most significant risk factor, accounting for approximately 80% of lung cancer cases. Other risk factors include exposure to environmental pollutants like radon, asbestos, and air pollution, as well as having a personal or family history of lung cancer [6,7].

Lung cancer is most commonly diagnosed in individuals over 65, with the median age at diagnosis being 70 years old. However, lung cancer can occur at any age, and there has been a recent increase in the incidence of lung cancer in younger individuals who have never smoked [1].

Despite significant advances in diagnosis and treatment, progress in lung cancer survival has been slow. Precision diagnosis and personalized treatment have become prominent in lung cancer care [8]. There have also been considerable advances in immunotherapy, radiotherapy, and surgical approaches. However, survival rates remain low. This is largely since most lung cancer patients are diagnosed at an advanced stage, with a five-year survival rate of only 13% (11.2% in men and 13.9% in women), with noticeable variations between Northern, Southern, Eastern, and Central Europe [9]. This unfavorable prognosis is associated with the delayed detection of the disease, as 50–70% of incident cases are typically diagnosed at an advanced stage (stage 4), whereas only 15–25% are detected at an early stage [9].

On the other hand, if lung cancer is detected at an early stage, 68% to 92% of people can survive more than five years. The early detection of lung cancer through screening programs can potentially reduce mortality rates and improve patient outcomes, particularly in high-risk individuals [10].

Lung cancer screening is a method of detecting lung cancer before symptoms appear through imaging tests. The goal of lung cancer screening is to detect the disease early, when it is easily treatable and potentially curable. The most commonly used technique for lung cancer screening is low-dose computed tomography (LDCT). This type of X-ray uses low doses of radiation to obtain detailed images of the lungs, and it has been shown to be effective in detecting lung cancer at an early stage when the cancer is still confined to the lungs and has not spread to other parts of the body [11].

In recent years, there has been increasing interest in lung cancer screening for earlier detection, which can lead to more effective treatment and improved survival rates.

In some countries, such as the United States, lung cancer screening programs using LDCT are offered in high-risk populations [12].

This practice aligns with the early diagnosis of cancer through screening, one of the top priorities of the European Cancer Plan 2021 [13]. Furthermore, in 2022, the Council of the European Union adopted a new document evaluating the feasibility of implementing lung cancer screening programs based on the latest scientific data [14].

In Italy, cancer screening programs provided by the National Health System (SSN) prioritize the early diagnosis of breast, cervical, and colorectal cancers without offering lung cancer screening [15].

However, Law No. 106 of 23 July 2021 authorizes the allocation of EUR one million to each center of the Italian Lung Cancer Screening Network to implement an LDCT program for high-risk individuals, as well as primary prevention interventions for supporting smoking cessation [16]. This significant investment suggests that implementing lung cancer screening programs may become the next challenge for public health [11].

Nevertheless, before organized screening programs can be established, the National Cancer Plan 2023–2027 emphasizes the importance of conducting risk stratification activities and evaluating the efficacy and effectiveness of such programs [17].

This study aims to provide a comprehensive systematic review of lung cancer screening, including its benefits, risks, and limitations. It also discusses the current state of lung cancer screening programs worldwide, evaluates the evidence for effectiveness and cost-effectiveness, and explores different implementation strategies. 

## 2. Materials and Methods

In October 2022, a systematic review was conducted using PubMed and Scopus as search engines. English-language studies published in scientific journals, letters to the editor, comments, and book chapters were included based on pre-determined criteria. The review was conducted following the Preferred Reporting Items for Systematic Reviews and Meta-Analyses (PRISMA) guidelines [18]. The search strategy was developed using the PICO framework [19], commonly used to formulate research questions and search strategies in evidence-based practice. To include pertinent literature on this topic, various search terms related to each section of the PICO framework were combined: population: lung cancer; intervention: screening; comparison: standard care; outcome: strategies. Using this PICO strategy, the following search string was used: (Population: “lung cancer”) AND (Intervention: “screening”) AND (Comparison: “standard care” OR “usual care” OR “no intervention”) AND (Outcome: “strateg*” OR “model*” OR “efficacy” OR “effectiveness”).

### Study Selection Criteria

Two reviewers (D.A., M.F.P.) critically evaluated each eligible study using a checklist. The checklist included criteria such as random assignment, comprehensive diagnostic work-up planning, adherence to inclusion criteria, accurate mortality measurement, and blinded outcome assessment. Data extraction was carried out independently by two double-blind re-examiners (D.A., M.F.P.). The extracted data included the article title, sample size, participant’s characteristics (age, gender, smoking history), type of intervention in the control group, median follow-up duration, number of screening rounds, time intervals between rounds, and number of deaths from all causes as well as those specifically related to lung cancer.

Through a title–abstract analysis, 297 articles were gathered and assessed. Those studies that did not pertain to the efficacy or effectiveness of lung cancer screening, or any strategies, approaches, or models used in it, were excluded. The remaining articles (n. 61) relevant to the criteria were selected for a thorough full-text content analysis. 

Figure 1 shows a flow diagram of the literature search strategy and the review process following PRISMA guidelines [18].

After the exclusion of 236 non-relevant articles, we included 61 full-text articles to critically evaluate. The number of papers broken down by year has been uploaded as Appendix A.

## 3. Results

### 3.1. Efficacy and Effectiveness Studies about Lung Cancer Screening

Lung cancer screening has been found to be effective in reducing mortality rates from lung cancer in certain high-risk populations, and its efficacy depends on various factors, including the type of screening test used, the frequency of testing, the age and smoking history of the individual being screened, and the availability of treatment options [20,21,22,23].

The most used screening method is LDCT, which uses a low radiation dose to create detailed images of the lungs. Over the years, several large randomized controlled trials have been conducted to evaluate the efficacy and effectiveness of lung cancer screening using LDCT.

In particular, the National Lung Screening Trial (NLST), the largest randomized controlled trial to date, was conducted in the United States to evaluate the efficacy of LDCT screening. This trial was conducted from 2002 to 2010 and included 53,454 current or former heavy smokers aged 55 to 74 who were randomly assigned to either LDCT screening or chest X-ray screening. The NLST divided the participants into two groups: a screening group that received three LDCT scans each year for three years and a control group that received only one chest X-ray each year for three years.

The results demonstrated that, after a median follow-up of 6.5 years, LDCT screening resulted in a 20% reduction in lung cancer mortality compared to chest X-ray screening (relative risk (RR) 0.80, 95% confidence interval (CI) 0.70–0.92). LDCT screening also exhibited higher sensitivity in detecting early-stage lung cancer cases than chest X-ray screening [24,25].

Based on these findings, in December 2013, the U.S. Preventive Services Task Force [26] and the Centers for Medicare and Medicaid Services (CMS) [27] recommended annual lung cancer screening for eligible high-risk individuals. This group includes individuals between the ages of 55 and 80 who have a smoking history equivalent to 30 packs of cigarettes per year and are currently smoking or have quit within the last 15 years. Pack-years is a measure of smoking intensity calculated by multiplying the number of packs of cigarettes smoked per day by the number of years of smoking [20,27].

While the implementation of LDCT lung cancer screening has been adopted in the United States due to conclusive evidence of reduced mortality, European healthcare systems have been cautious about issuing similar recommendations. This hesitation stems from early analyses of European clinical trials, which were constrained by limited statistical power in confirming differences in mortality between LDCT and control groups [28].

Furthermore, European trials exhibited heterogeneity in the criteria for cohort selection, types of interventions used in the control group, screening frequency, definition of positive screening, and nodal management strategies. These variations make direct comparisons and data pooling challenging [28].

Among European studies, the Dutch–Belgian Randomized Lung Cancer Screening Trial (NELSON) was conducted in the Netherlands and Belgium to evaluate the efficacy of LDCT lung screening. This randomized controlled trial, the only European trial with adequate power, began in 2003 and completed its final follow-up in 2019. Participants were randomly assigned to a screening group that received four LDCT scans every two years for four years or a control group that received no screening. The study compared LDCT screening to no screening in 15,792 individuals aged 50 to 74 who were current or former heavy smokers. The results showed that LDCT screening led to a 24% (RR 0.76, 95% CI: 0.61–0.94) reduction in lung cancer deaths in asymptomatic men at a high risk for lung cancer after ten years of follow-up. This trial also found that false-positive results decreased over time as radiologists gained experience interpreting the scans [29]. 

Several smaller randomized controlled trials have also evaluated the efficacy of LDCT screening for lung cancer, including the Danish Lung Cancer Screening Trial and the Italian Lung Screening Trials.

The Danish Lung Cancer Screening Trial (DLCST) was a randomized controlled trial conducted in Denmark, which compared LDCT screening versus no screening in a group of 4104 individuals aged 50 to 70 years who were at a high risk for lung cancer (current or past smokers with a smoking history of at least 20 pack-years). The participants were randomly assigned to receive five annual LDCT scans or no screening. The trial began in 2004, and the final follow-up was completed in 2014; no statistically significant effect of CT screening on lung cancer mortality was found (RR 1.03; 95% CI 0.66–1.60). Post hoc high-risk subgroup analyses showed a nonsignificant trend, which appears to be in good agreement with the results of the National Lung Screening Trial (NLST). In this trial, screening detected more early-stage lung cancer and less late-stage lung cancer, indicating that screening may contribute to the earlier detection and treatment of lung cancer [30,31,32,33,34].

In Italy, one of the most important efficacy studies on lung cancer screening was the Italian Lung (ITALUNG) Screening Trial. This randomized controlled trial compared LDCT screening to no screening in 3206 individuals at a high risk for lung cancer aged 55 to 69 years. The trial found that LDCT screening detected more early-stage cancers than no screening, with a reduction in lung cancer mortality by 30% (RR = 0.71; 95% CI 0.48 to 1.04) compared to no screening. However, the reduction in mortality was not statistically significant due to the relatively small sample size of the trial [33].

Another study conducted in Italy from 2005 to 2018 was the Multicentric Italian Lung Detection (MILD) trial, which investigated the efficacy of lung cancer screening using LDCT. The study enrolled 4099 asymptomatic individuals aged 49 to 75 years with a smoking history of at least 20 pack-years. The main results of the MILD study were that LDCT screening reduced lung cancer mortality by 39% at ten years compared to no screening (RR 0.61; 95% CI 0.39–0.95). The MILD findings have contributed to the growing body of evidence supporting using LDCT for lung cancer screening. Like the other lung cancer screening trials, the MILD study also found a high rate of false-positive results, which can lead to unnecessary further testing and potential harm to the patient.

Overall, the MILD study provides further support for the efficacy of LDCT screening for lung cancer. However, it also highlights the importance of the careful consideration of the potential harms and benefits of screening for each individual [34,35].

The DANTE Italian trial enrolled 2472 current or former heavy smokers aged 60 to 74 years. The DANTE study did not provide conclusive evidence that LDCT screening reduces lung cancer mortality in high-risk individuals (RR 0.99, 95% CI 0.69–1.43). However, the study showed a trend toward reducing lung cancer mortality and confirmed that LDCT screening can detect more early-stage lung cancers. This is consistent with the results of other LDCT screening trials. The study also highlights the potential harms of LDCT screening, including a high rate of false-positives and overdiagnosis [36].

Another important study, the UKLS (UK Lung Cancer Screening), conducted in the United Kingdom, was a randomized controlled trial that compared LDCT screening to no screening in over 4055 individuals aged 50–75 at a high risk for lung cancer. The study found that LDCT screening detected more early-stage cancers than no screening, but it did not find a significant reduction in lung cancer mortality [37].

The randomized German LUSI trial (Lung Cancer Screening Intervention) was conducted to investigate the effectiveness of LDCT screening in reducing lung cancer mortality. The trial was carried out between 2007 and 2011 and involved 4052 participants between the ages of 50 and 69 years who were current or former smokers with a smoking history of at least 20 pack-years. The trial results showed that LDCT screening reduced lung cancer mortality by 26% (HR 0.74; 95% CI 0.46–1.19, *p* = 0.21) compared to no screening. However, modeling by gender showed a statistically significant reduction in lung cancer mortality in women screened by LDCT (HR = 0.31 (95% CI: 0.10–0.96), *p* = 0.04), but not in men (HR = 0.94 (95% CI: 0.54–1.61), *p* = 0.81) screened by LDCT. However, this study also found that screening with LDCT leads to a high rate of false-positives and the overdiagnosis of lung cancer, as reported in several trials [38].

The paper by Blanchon et al. [39] reports the baseline results of the DEPISCAN trial, a French randomized pilot lung cancer screening trial comparing LDCT and chest X-ray (CXR). The study enrolled 765 participants between the ages of 50 and 75 who were current or former heavy smokers.

The authors reported that the prevalence of non-calcified lung nodules was higher in the LDCT group compared to that in the CXR group. Specifically, 45.2% of participants in the LDCT group had nodules detected, compared to 7.4% in the CXR group, and most nodules were small (less than 5 mm in diameter) [39].

Although the trials did not have enough statistical power to establish a significant impact on reducing overall mortality, modeling analyses indicated that, on average, each lung cancer death prevented could potentially result in a gain of 10 life-years after adjusting for the increased mortality from other causes among smokers and former smokers [40].

Until now, only a limited number of studies have examined the effectiveness of CT screening for lung cancer in non-smokers or those with a history of light smoking. For example, the Hitachi study was an ecological/time series study that evaluated the effectiveness of a lung cancer screening program using LDCT in the Hitachi district of Japan. The study included both smokers and non-smokers, with non-smokers accounting for approximately half of the CT screening examinees. The study found that the lung cancer detection rate in non-smokers was lower than that in smokers but was still significant. Specifically, the lung cancer detection rate was 0.33% in non-smokers and 1.27% in smokers. 

The study also found that the stage distribution of lung cancer in non-smokers was more favorable than that in smokers, with a higher proportion of early-stage lung cancer detected in non-smokers. Overall, the study suggested that CT screening may effectively detect early-stage lung cancer in non-smokers and light smokers. However, more research is needed to confirm these findings and to determine the optimal screening criteria for these populations [41,42].

A summary of lung cancer screening trials is reported in Table 1.

### 3.2. Risks and Benefits of LDCT Lung Cancer Screening

LDCT lung cancer screening has both risks and benefits. One of the most significant advantages of this screening test is its ability to detect lung cancer early. When lung cancer is detected early, it can be treated more effectively, and the chances of survival are greatly increased. Furthermore, LDCT lung cancer screening is a non-invasive procedure, which means no need for a biopsy or other invasive diagnostic tests. This makes the screening process much more comfortable for patients. The screening test is quick and painless, usually taking less than 10 min to complete. Finally, LDCT lung cancer screening is relatively cost-effective compared to other screening tests, such as magnetic resonance imaging (MRI) or positron emission tomography (PET) scans, making it a more accessible option for patients [43].

Despite the benefits of lung cancer screening, there are also limitations to consider. Screening is typically recommended for individuals at a high risk of developing lung cancer, such as those with a smoking history. However, not all individuals at a high risk of developing lung cancer will benefit from screening. For example, individuals with a history of lung cancer or those unable to undergo surgery may not benefit from screening [43,44]. Patients unfit for surgery may still benefit from lung cancer screening programs, as they can be offered alternative treatments such as stereotactic ablative radiotherapy (SABR), also known as stereotactic body radiation therapy (SBRT). SABR is a non-invasive treatment option that delivers high doses of radiation precisely to the tumor while minimizing damage to surrounding healthy tissues. It has shown promising outcomes in the treatment of early-stage non-small-cell lung cancer (NSCLC) and can be a suitable alternative for patients who are not candidates for surgery [43].

Another factor that can impact the efficacy of lung cancer screening is the frequency of screening. The optimal screening interval needs to be better defined, and the frequency of screening may vary depending on individual risk factors and other factors such as age and comorbidities [24,43].

Furthermore, false-positives are a significant concern, as LDCT can sometimes detect non-cancerous nodules or tumors, leading to unnecessary invasive procedures such as biopsies, bronchoscopy, or lung surgery that may have adverse effects on an individual’s quality of life and result in higher healthcare expenses and anxiety for the patient. However, nodules or tumors that are not cancerous require a structured and comprehensive healthcare approach with evaluation and management through the interprofessional team in caring for patients with this condition. 

False-negatives are also a risk, as not all lung cancers may be detected by LDCT [24,43,45]. 

According to the NLST findings, 39.1% of the participants had received at least one positive result out of the three annual CT screens. Out of those, 96.4% were discovered to be false-positives, and 72.1% of those false-positives required further (invasive) investigations referred by a pulmonologist. In total, 23.3% of all CT screens in NLST were false-positives. Only 3.6% of the screens led to lung cancer diagnosis, and 2.7% of the participants who had false-positive screening results faced complications after undergoing the investigations [22,24,25].

In the NELSON study, 59.4% of participants who had an initial positive result during four screening rounds were found to be false-positives, resulting in an overall low false-positive rate of 1.2% [46,47].

Another limitation is the potential for overdiagnosis, overtreatment, and radiation exposure. Overdiagnosis occurs when patients are diagnosed with lung cancer that would not have caused symptoms or threatened their life, leading to unnecessary treatment and potential harm. Radiation exposure from LDCT screening can increase the risk of cancer, although the risk is low compared to the potential benefits of screening.

Finally, the cost of lung cancer screening is a consideration, as it can be expensive, and insurance coverage for screening may vary. This can make screening inaccessible to some patients who may benefit from it. In addition, the costs of follow-up testing and treatment can add up quickly, especially if a false-positive result leads to multiple diagnostic tests or procedures. [43].

### 3.3. The Most Currently Used Approaches for Lung Cancer Screening 

Many countries have implemented screening programs for lung cancer early detection, but screening recommendations for lung cancer may vary depending on the available resources and infrastructure (Table 2). The approaches currently used for lung cancer screening include LDCT, chest X-rays, sputum cytology, biomarker tests, and risk assessment tool rates [48].

LDCT is currently the most widely recommended approach for lung cancer screening in many countries and by several organizations, including the United States Preventive Services Task Force (USPSTF) and the American Cancer Society (ACS). LDCT has been shown to be more effective than chest X-rays in detecting early-stage lung cancer and reducing lung cancer mortality. The advantage of the LDCT-based protocol is its simplicity and its high sensitivity. Refined criteria defining positive findings that are largely based on the nodule size and/or volume reduce false-positive rates [48]. 

In particular, in the United States, screening for lung cancer is recommended using LDCT. The USPSTF recommends LDCT screening for individuals at a high risk of developing lung cancer, particularly those aged 50 to 80 who have smoked at least an average of one pack of cigarettes per day for at least 30 years and currently smoke or have quit within the past 15 years [45].

In Canada, most guidelines recommend LDCT screening for individuals at a high risk of developing lung cancer, particularly those aged 55 to 74 who have smoked for at least 30 years [49].

In the UK, the National Health Service (NHS) offers screening for lung cancer through individual risk assessment based on a questionnaire and assessment of current health conditions. If a high risk is identified, an LDCT screening is offered [50].

In Australia, screening for lung cancer is primarily carried out through LDCT, mainly for high-risk individuals. Specifically, screening is recommended for individuals aged 55 to 74 who have smoked at least an average of one pack of cigarettes per day for at least 20 years [51].

Furthermore, based on the results of the NELSON study, the Netherlands has implemented a national lung cancer screening program. The program targets individuals who are between 50 and 75 years old, have a history of smoking, and have smoked at least 15 cigarettes per day for at least 25 years or at least 10 cigarettes per day for at least 30 years. Participants in the program receive an LDCT scan every year for three years and then every two years for the next four years if the initial scans are negative.

The Dutch lung cancer screening program also includes smoking cessation counseling for current smokers and follow-up testing for individuals who are found to have suspicious nodules or masses on their LDCT scans. The program is expected to reduce mortality from lung cancer and improve the health outcomes of individuals who participate.

In Japan, screening for lung cancer is primarily carried out through chest X-rays, but LDCT is becoming increasingly common. Screening is offered to individuals aged 40 to 74 who have smoked for at least 20 years [52].

South Korea has a lung cancer screening program with LDCT for high-risk people between 55 and 74 [53].

In China, screening for lung cancer is primarily carried out through LDCT. However, screening programs vary depending on the region, city, and population groups. For instance, in 2015, the Chinese government launched a pilot screening program for lung cancer in Shanghai, which provided free LDCT to residents at a high risk of lung cancer. In other parts of the country, screening programs have primarily been initiated in hospital or clinic settings, offering LDCT to high-risk individuals.

Overall, in China, lung cancer screening is still relatively limited. However, it is becoming increasingly relevant, especially with the growing awareness of the risks of smoking and the need for the early diagnosis of lung cancer [54].

Another approach is chest X-rays, which were previously used for lung cancer screening, but their effectiveness in detecting early-stage lung cancer has been questioned in recent years. Multiple studies have shown that chest X-rays are less effective than LDCT for lung cancer screening. This is because chest X-rays may miss small lung nodules that could indicate early-stage lung cancer. In contrast, LDCT can detect smaller nodules than chest X-rays and has been shown to improve the detection of lung cancer at an earlier stage when it is more treatable [55].

However, chest X-rays may still have a role in certain situations, such as for individuals who cannot undergo CT scans for medical reasons or for those living in areas where LDCT is not readily available. Additionally, chest X-rays may be used as a follow-up test for individuals with a positive LDCT result [56].

Also, another approach is sputum cytology, which involves analyzing sputum samples for abnormal cells that could indicate the presence of lung cancer. However, it is not recommended as a standalone screening tool because of its limited sensitivity and specificity. Sputum cytology is mostly used as a complementary test to LDCT [56].

Finally, some countries, such as China, have developed biomarker tests for detecting lung cancer by analyzing a patient’s blood or other bodily fluids for specific proteins or genetic markers. Biomarker tests, such as those that detect specific proteins or genetic mutations in the blood, are not currently recommended as a screening tool for lung cancer. At the same time, these tests may have some value in guiding treatment decisions for patients with lung cancer [57].

Lastly, some countries, such as Australia, use risk assessment tools to determine which individuals are most likely to develop lung cancer and should be screened. These tools consider factors such as age, smoking history, and family history of lung cancer. Risk assessment tools, such as the Lung Cancer Risk Assessment Tool (PLCOm2012) or the Liverpool Lung Project Risk Model, can help identify individuals at an increased risk for developing lung cancer. However, these tools are not used as a screening tool in and of themselves; they are used to identify individuals who may benefit from further screening with LDCT.

### 3.4. Lung Cancer Risk Prediction Models

Lung cancer risk prediction models are statistical tools that use a set of known risk factors (such as age, smoking history, family history of lung cancer, and other relevant factors) to estimate an individual’s probability of developing lung cancer within a given period. These models are developed using data from large studies and aim to identify individuals at a high risk for lung cancer so that they can be targeted for early detection and preventive measures. The models can be used in clinical practice to help inform decision making around screening and risk-reducing interventions.

There are several lung cancer risk prediction models, some of which are based on known risk factors for the disease, including:

The National Lung Screening Trial (NLST) risk prediction: this model was developed using data from the National Lung Screening Trial and is based on a person’s age, smoking history, and history of chronic obstructive pulmonary disease (COPD). The model calculates a person’s risk of developing lung cancer over 6 years [58].

The Liverpool Lung Project (LLP) risk prediction: this model uses a person’s age, smoking history, family history of lung cancer, and occupational exposure to carcinogens to predict their risk of developing lung cancer over 5 years [59,60].

The Prostate, Lung, Colorectal, and Ovarian (PLCO) Cancer Screening Trial risk prediction: this model uses a person’s age, smoking history, family history of lung cancer, and history of pneumonia to predict their risk of developing lung cancer over 6 years. The model was developed using data from the PLCO Cancer Screening Trial, a large randomized controlled trial of cancer screening [61,62].

The International Early Lung Cancer Action Program (I-ELCAP) risk prediction: this model uses a person’s age, smoking history, and history of lung disease to predict their risk of developing lung cancer over 5 years. The model was developed using data from the International Early Lung Cancer Action Program, a large screening study [63,64].

The Bach model is a lung cancer risk prediction model developed by Bach et al. It is designed to estimate an individual’s risk of developing lung cancer based on several key factors. The model considers variables such as age, smoking history, family history of lung cancer, history of chronic obstructive pulmonary disease (COPD), and history of previous cancer. The Bach model predicts the risk of developing lung cancer over a specified period of 5 years [65].

The American Cancer Society (ACS) lung cancer risk prediction: this model uses a person’s age, smoking history, and history of COPD to predict their risk of developing lung cancer over 6 years. The model was developed using data from the Cancer Prevention Study II Nutrition Cohort, a large study of cancer risk factors [66,67].

All these models use statistical methods to estimate a person’s risk of developing lung cancer based on their risk factors. The models vary in the specific risk factors they include and the length of time over which they predict risk (Table 3).

### 3.5. Cost-Effectiveness Studies

The cost-effectiveness of lung screening was investigated in several studies where the input parameters and models were different, as well as the setting and target groups and related results (Table 4).

Here, we briefly present the manuscripts identified in the present research.

The benefits of screening by comparing the absolute and relative difference in lung cancer-specific deaths was performed by a group of scholars who measured harms by the number of false-positive invasive tests or surgeries per 100,000 and the incremental cost-effectiveness in U.S. dollars per quality-adjusted life-year (QALY) gained, using a computer-simulated model [68].

Mahadevia PJ et al. found an incremental cost-effectiveness for current smokers of USD 116,300 per QALY gained; lower ICERs was detected in quitting and former smokers, with values of USD 558,600 and USD 2,322,700 per QALY gained, respectively. Different parameters could influence the analysis, such as the degree of the stage shift, compliance with screening, and cost of the test used [68].

Pinsky PF [82] argued that Black et al. estimated the cost per QALY of low-dose CT screening for subgroups enrolled in the NLST. Gender differences were revealed—particularly, the cost per QALY gained was triple for men with respect to women, probably due to the observed difference in the reduction in lung cancer mortality associated with low-dose CT screening (8% in men vs. 27% in women).

In another study, the QALYs, costs per person, and ICERs for three alternative strategies: screening with low-dose CT, radiography, and no screening [69] were calculated. Specifically, lung screening by means radiography did not give health benefits compared to no screening; screening with low-dose CT was more costly (an additional USD 1631 per person) and able to provide an added 0.0316 life-years per person (95% CI, 0.0154 to 0.0478) and 0.0201 QALYs per person (95% CI, 0.0088 to 0.0314). The ICERs were under the threshold of USD 100,000 per QALY gained—specifically, USD 81,000 per QALY gained and USD 52,000 per life-year gained.

Toumazis I. et al. performed three studies on the cost-effectiveness analysis of lung cancer screening with low-dose computed tomography [70,71,72].

In the most recent study [70], the authors performed a simulation of a cohort consisting of 1 million individuals, targets of the U.S. lung cancer screening, and this cohort was followed from age 50 for a period of 45 years. 

The model showed that personal risk-based screening for lung cancer was cost-effective under a wide range of risk thresholds, exhibiting flexibility for implementing risk model-based approaches in a variety of settings; the strategy with a 6-year risk threshold of 1.2% or superior was cost-effective, with an ICER less than USD 100,000 per QALY and an exact ICER of USD 94,659.

Analogously, previous research displayed positive findings for a lung cancer screening program following the U.S. Preventive Services Task Force (USPSTF) guidelines and using a highly diagnostic sensitive and specificity biomarker with a price of USD 250 or less, resulting in cost-effectiveness and improving lung cancer-specific mortality reduction [71]. 

Another study by Toumazis I. et al. [72] focused on the performance of the USPSTF recommendations on lung cancer screening and found that the lung cancer screening program was cost-effective. In particular, the 2021 U.S. Preventive Services Task Force recommendation showed greater cost-effectiveness (mean ICER USD 66,533).

A very recent updated Australian evaluation for lung cancer screening assessed its cost-effectiveness. The scholars applied screening parameters and outcomes from NLST and the NELSON to Australian data on lung cancer [73]; the ICERs resulted in AUD 39,250 (95% CI AUD 18,150–108,300) per QALY, yielding more benefits with respect to the previous evaluations. 

In order to compare the cost-effectiveness of different stopping ages for lung cancer screening, four microsimulation models assessed the health and cost outcomes of annual lung cancer screening with LDCT in U.S. [74]. The ICERs found were USD 49,200, USD 68,600, and USD 96,700 per QALY, respectively, for the NLST, CMS, and USPSTF screening strategies.

The cost-effectiveness of the implementation of a national lung cancer screening program in a high-risk population from the perspective of the Spanish National Health System (NHS) was investigated by Gómez-Carballo N et al. [75]. In summary, the study compared two scenarios: one with the implementation of the screening program and another without the implementation of lung cancer screening in the high-risk Spanish cohort, which was cost-effective, as an increase of 4.80 QALYs per patient and an incremental cost-effectiveness ratio of EUR 2345/QALY were found.

Kumar V. and colleagues used data from the NLST for a multistate model to predict health state transitions [76]. Health benefits were demonstrated, highlighting a prevention of 1.2 deaths per 10,000 person-years compared to 9.5 deaths from lung cancer per 10,000 person-years for patients at the highest risk. The ICER for LDCT was USD 37,000/LY gained or USD 60,000/QALY gained in the overall NLST cohort.

Hinde S. et al. performed an economic evaluation using the methodological approach reported in the UKLS trial to estimate the cost-effectiveness of LDCT for lung cancer. Assessments of the total costs and quality-adjusted life-years (QALYs) were calculated [77], finding an ICER of GBP 10,069/QALY, resulting in cost-effectiveness using controlled NHS resources. A cost-effectiveness analysis considering the public payer perspective for a high-risk population (heavy former and current smokers aged 55–75 years) was conducted by German researchers using two Markov models. These models evaluated a population-based annual screening program compared to standard clinical care [78].

The outcomes measured were costs, life-years saved, and QALYs, with input parameters taken from the literature. 

The base case analysis showed that annual lung cancer screening increased incremental costs (EUR 1153 per person) compared to standard clinical care. The screening approach was associated with an incremental gain in life-years (0.06 per person) and QALYs (0.04 per person). The ICER was EUR 19,302 per life-year saved and EUR 30,291 per QALY. The study concluded that lung cancer screening for a high-risk population might be more effective, but also more costly, than standard clinical care from the perspective of a German payer.

A recent Chinese study by Zhao Z. investigated the cost-effectiveness of lung cancer screening among heavy smokers by incorporating the start age and screening interval [79]. The scholars revealed that a lung cancer screening program in China for heavy smokers using low-dose computed tomography was cost-effective for individuals with smoking habits aged 55–74 years, along with one-time screening for those aged 65–74 years. 

The authors found that mortality reduction due to lung cancer ranged from 0.004% to 1.171% for one-time screening and from 6.189% to 15.819% for annual screening. The ICER ranged from CNY 119,974.08 to CNY 614,167.75 per QALY gained compared to that of non-screening. According to the World Health Organization threshold of CNY 212,676 per QALY gained, annual screening starting at 55 years and one-time screening starting at 65 years can be considered cost-effective in China. Based on these findings, the authors argued in favor of promoting annual lung cancer screening to realize the benefits of a recommended screening program.

The cost-effectiveness and health impact of lung cancer screening with LDCT for never-smokers were investigated in two different contexts: Japan and the U.S. [80]. The evaluation highlighted that screening was cost-saving in Japan, while it was not cost-effective in the U.S. Particularly, LDCT for 60-year-old Japanese people allowed for substantial savings of about USD 117 billion, and 224,749 deaths could be prevented.

Finally, an Italian research group used a decision model in order to evaluate the cost-effectiveness of annual screening for five years in high-risk smokers (aged 55–79 with ≥30 pack-years) [81].

The study results revealed an ICER of EUR 3297 per quality-adjusted life-year and EUR 2944 per life-year gained in the base case. The findings suggested that low-dose computed tomographic screening could be introduced in Italy at a reasonable cost, saving the lives of many lung cancer patients. The estimated cost-effectiveness ratios for screening compared to usual care in Italy were lower than the acceptable thresholds set by the UK’s National Institute for Health and Care Excellence (NICE) for introducing new medical technologies [83].

### 3.6. The Importance of Tobacco Cessation

Tobacco is a known cause of various types of tumors, accounting for approximately 85% of lung cancer cases and 30% of cancer mortality [84].

Optimizing smoking cessation services within an LDCT lung cancer screening program has the potential to improve the cost-effectiveness and the overall efficacy [85].

However, there is limited evidence regarding the optimal design and integration of tobacco cessation services.

A personalized intervention booklet, utilizing LDCT scan images, has been developed for delivery by trained smoking cessation practitioners. The results highlight the benefits of co-development during intervention creation and emphasize the need for further evaluating its effectiveness [86].

In a study by Park et al. [86], the impact of counseling using the 5As approach (Ask about smoking, Advise to quit, Assess readiness to quit, Assist with tobacco dependence treatment, and Arrange follow-up) on smoking cessation was evaluated. The study focused on a subset of smokers enrolled in the National Lung Screening Trial (NLST). The results showed that the “assist” and “arrange” steps of 5As counseling were associated with increased odds of quitting at 12 months.

In a study by Bade et al. [87], smoking cessation rates were compared between patients who underwent LDCT screening and those who did not. The study revealed higher smoking cessation rates among patients who attended counseling sessions in the LDCT screening group at 12 months (14.6%) and 24 months (12.9%) compared to those who did not attend counseling (12 months: 6.7%, 24 months: 7.6%, *p*-values from the analysis: *p* < 0.0001 and *p* = 0.002, respectively).

In order to be effective, lung cancer screening requires a multidisciplinary approach, encompassing individualized risk assessment, shared decision making, smoking cessation, structured reporting, high-quality multi-specialty cancer care, and reliable follow-up. Specialized organizations have outlined the key components and metrics that screening programs should incorporate. Ongoing research focuses on long-term outcomes, the refinement of screening criteria, and the use of biomarkers for early cancer detection [88].

A quasi-experimental study by Luh et al. examined a screening program emphasizing primary prevention by encouraging smoking cessation [89]. The study found that patients who received counseling from physicians and nurses showed greater odds of advancing in terms of readiness to quit compared to a control group (OR 2.27, 95% CI 1.07–4.84), while patients who received a smoking cessation leaflet had no significant difference (OR 0.99, 95% CI 0.44–2.25). [90].

Zeliadt et al. conducted a pilot feasibility trial to evaluate the impact of proactive outreach telephone counseling on behavioral cessation support and quit rates [90]. The study showed that patients who received the intervention had higher rates of using behavioral cessation support programs than the control group (44% vs. 11%, RR 4.1, 95% CI 1.7–9.9). 

Lung cancer screening could prompt current smokers to reflect on their health and might present an opportunity to engage them in discussions about smoking cessation. 

## 4. Discussion

Our systematic review on lung cancer screening using LDCT explored the benefits, risks, limitations, and current state of lung cancer screening programs in several countries. The evidence of the effectiveness and cost-effectiveness of lung cancer screening was also investigated. 

The results highlighted that lung cancer screening has been proven effective in reducing mortality rates due to lung cancer in certain high-risk populations, and its efficacy depended on various factors, such as the type of screening test used, the frequency of testing, the age and smoking history of the individual being screened, and the availability of treatment options. However, the lack of a significant reduction in lung cancer mortality, despite the detection of lung cancer cases through screening, has been observed in some studies [30,31,32,33,34,36,37]. It is important to consider that individual studies may have limitations that could affect the observed results in terms of mortality. Variations in the study design, participant characteristics, follow-up duration, overdiagnosis, and smoking habits can contribute to differences in mortality outcomes. Additionally, the relatively short follow-up duration in some studies may not capture long-term mortality benefits that could emerge over a longer period.

Furthermore, most of the research conducted primarily focused on the population at a high risk of developing lung cancer, and only a limited number of studies focused on the general population’s health [91].

One major issue regarding lung cancer risk and screening is defining target populations at risk. This emphasis on high-risk individuals is exemplified by studies such as the National Lung Screening Trial (NLST), which included older participants with a significant smoking history. Subgroup analyses within NLST divided the population into quintiles according to the risk of lung cancer–associated death, and lung cancer-specific mortality increased significantly with each quintile.

In addition, to evaluate the effectiveness of lung cancer screening, the review included a cost-effectiveness analysis. Most of the examined studies report favorable incremental cost-effectiveness ratios (ICERs) below the threshold of EUR 100,000. This finding indicates that lung cancer screening using LDCT is cost-effective in most cases. However, it is important to note that one study found a value indicating a lack of cost-effectiveness.

Several countries, such as the U.S., have begun implementing lung cancer screening, and the offer is addressed to high-risk individuals.

However, the recent European recommendations regarding screening programs’ implementation emphasize the need to evaluate the optimal strategies for selecting and recruiting the population, the management of the results, and the integration of smoking cessation programs. A limited personnel resource capacity could be a potential barrier, as well as investments in infrastructure.

Although LDCT cancer screening has shown its efficacy in reducing lung cancer mortality, its implementation is complex and requires specific and multidisciplinary skills. Indeed, it is essential to carefully consider the potential risks and benefits for each individual and to tailor interventions to the subpopulations with the highest non-adherence rates due to low risk perception.

In order to minimize the risk of false-positives and -negatives, it would be relevant to include special software/tools for the interpretation of the exams and undergo special training for radiologists.

Indeed, false-positives and overdiagnoses not only impose economic costs on society but also cause unnecessary anxiety and worry in patients, with additional risks linked to the accumulation of radiation, often invasive diagnostic tests, and the surgical resection of a lobe or of the entire lung. All factors should be weighed well, and even individuals who want to undergo the exam individually should be duly informed. This is impossible in the current state of knowledge, in which these elements have not yet been quantified precisely. 

In addition to the aspects mentioned above, there are several other important factors to consider when implementing lung cancer screening programs using LDCT. These factors include ethical implications, equitable access, patient education, quality assurance, and the role of healthcare providers.

The implementation of lung cancer screening raises important ethical concerns. It is essential to ensure that individuals receive clear and comprehensive information about the benefits, risks, and limitations of screening. Informed consent should be obtained from participants, and privacy and confidentiality should be maintained throughout the screening process. Additionally, there should be mechanisms for addressing any potential conflicts of interest among healthcare providers and researchers involved in the screening programs [91].

To ensure equitable access to lung cancer screening, it is crucial to identify and address any existing disparities in healthcare access. Efforts should be made to reach populations at a high risk of developing lung cancer, including those from socioeconomically disadvantaged backgrounds and marginalized communities. Strategies such as community outreach, mobile screening units, and targeted awareness campaigns can help improve access to screening for underserved populations [92].

Comprehensive patient education is essential to promoting shared decision making in lung cancer screening. Patients should be provided with clear and understandable information about the benefits and risks of screening, as well as the potential consequences of false-positives and overdiagnosis. This education should also emphasize the importance of smoking cessation and integrating cessation programs with screening initiatives. Engaging patients in shared decision making empowers them to make informed choices based on their preferences and values [93].

Quality assurance measures play a critical role in ensuring the accuracy and reliability of lung cancer screening programs. These measures include standardized protocols for LDCT imaging, the regular calibration and maintenance of equipment, and ongoing training and certification for radiologists and other healthcare professionals involved in the interpretation of screening results. Quality assurance programs should also incorporate mechanisms for monitoring and evaluating the performance of screening centers, including the assessment of false-positive and false-negative rates [94]. 

Healthcare providers play a crucial role in the success of lung cancer screening programs. They need to be knowledgeable about the benefits and limitations of LDCT screening and effectively communicate this information to their patients. Healthcare providers should also be well informed about the appropriate referral pathways for individuals with abnormal screening results and facilitate timely follow-up and treatment for those diagnosed with lung cancer [90].

Collaboration among researchers, policymakers, healthcare providers, and patient advocacy groups is essential to optimizing the implementation of lung cancer screening programs. Continued research is needed to further refine the screening guidelines, identify high-risk populations, develop risk prediction models, and evaluate the long-term outcomes of screening. Additionally, research should focus on integrating emerging technologies, such as artificial intelligence and machine learning, to improve the accuracy and efficiency of LDCT screening. The integration of data mining, deep learning, and other emerging technologies, along with other innovative approaches, can indeed enhance the effectiveness and efficiency of lung cancer screening programs. Future research can focus on exploring the potential of these new technologies in optimizing lung cancer screening protocols, refining risk assessment models, and developing computer-aided diagnostic tools for improved patient outcomes [91].

## 5. Conclusions

Lung cancer is the leading cause of cancer-related deaths worldwide, and early detection is crucial to improving patient prognosis. Lung cancer screening with LDCT scans has been shown to reduce mortality from lung cancer, but it also carries potential risks, such as false-positives and overdiagnosis. Quality assurance ensures accurate and reliable lung cancer screening through standardized protocols, equipment maintenance, professional training, and the monitoring of screening center performance for false-positive and false-negative rates. Guidelines have been developed to help identify which individuals would benefit most from screening; however, it is important to discuss the risks and benefits of screening with a healthcare provider to make an informed decision. Ongoing research and technological advances will hopefully make lung cancer screening even more effective in the future.

## Figures and Tables

**Figure 1 healthcare-11-02085-f001:**
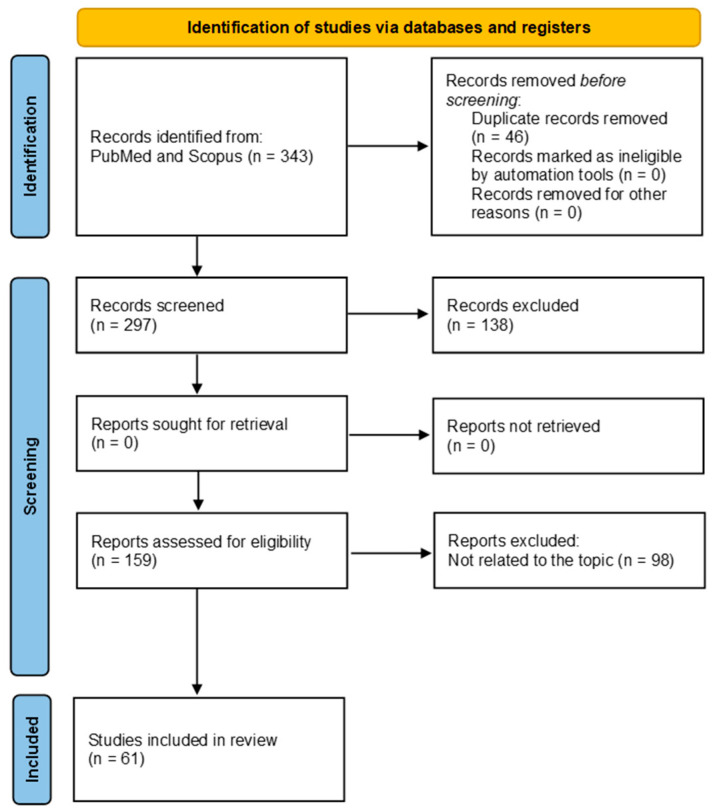
Flow diagram of the literature search strategy and review process, following PRISMA flow diagram rules.

**Table 1 healthcare-11-02085-t001:** Summary of lung cancer screening trials.

Study	Screening Method	Study Design	Country	Recruitment Period	Participants	Age (Years)	Ex Smokers (Years) Pack-Year	Main Findings	Ranking *
National Lung Screening Trial (NLST) [24,25]	LDCT vs. Chest X-ray	Randomized controlled trial	United States	2002–2004	53,454 current or former smokers	55–74	≥30, <15	Significant 20% reduction in lung cancer mortality among high-risk individuals compared to that from chest X-ray. Increased detection of early-stage lung cancer.	1
Dutch–Belgian Randomized Lung Cancer Screening Trial (NELSON) [29]	LDCT vs. usual care	Randomized controlled trial	The Netherlands	2003–2006	15,792 current or former smokers	50–74	≥10, ≤10	Significant 24% reduction in lung cancer mortality in the LDCT group compared to that in the control group.	2
Danish Lung Cancer Screening Trial (DLCST) [30,31,32,33,34]	LDCT vs. usual care	Randomized controlled trial	Denmark	2004–2006	4104 current or former smokers	50–70	≥20, <10	No significant differences in lung cancer or all-cause mortality were observed between the screening and control groups. More cancers were detected in the screening group, particularly adenocarcinomas, and at earlier stages.	5
Italian Lung Cancer Screening Trial (ITALUNG) [33]	LDCT vs. usual care	Randomized controlled trial	Italy	2004–2006	3206	55–69	≥20, <10	No significant difference in lung cancer mortality between the intervention and control groups.	7
Multicentric Italian Lung Detection (MILD) trial [34,35]	LDCT vs. usual care	Randomized controlled trial	Italy	2005–2018	4099 current or former smokers	49–75	≥20, <10	LDCT screening showed a 39% lower risk of lung cancer mortality at 10 years and a 20% reduction in overall mortality.	3
DANTE trial [36]	LDCT vs. usual care	Randomized controlled trial	Italy	2001–2006	2472 current or former smokers	60–74	≥20, <10	LDCT screening not conclusive for reducing lung cancer mortality. Trend towards reduction and early-stage detection. High false-positives and overdiagnosis.	8
UK Lung Cancer Screening Trial (UKLS) [37]	LDCT vs. usual care	Randomized controlled trial	United Kingdom	2011–2014	4055	50–75	Target population with a 5% risk of developing lung cancer within five years.	Higher detection of early-stage lung cancers, but final mortality reduction data still awaited.	6
LUSI trial [38]	LDCT vs. usual care	Randomized controlled trial	Germany	2007–2011	4052	50–69	>15, <10	Improved detection of lung cancer, higher rates of curative-intent treatment, and reduced mortality by 26% compared to no screening.	4
DEPISCAN trial [39]	LDCT vs. Chest X-ray	Randomized pilot trial	France	2002–2004	765 current or former smokers	50–75	≥15, <15	LDCT detected more non-calcified lung nodules than CXR. The majority were small (less than 5 mm)	9

* The studies were ranked based on the potential impact on early diagnosis and survival rates, considering the main findings of each trial. The ranking was subjective and based on the available information provided in the main findings of the studies. The studies with significant reductions in lung cancer mortality and a higher detection of early-stage lung cancer were ranked higher (e.g., NLST and NELSON). Studies that did not show significant reductions in lung cancer mortality but demonstrated improved detection or trends towards reduction were ranked lower (e.g., DANTE and UKLS). The DEPISCAN trial, which focused on detecting lung nodules, was ranked last.

**Table 2 healthcare-11-02085-t002:** Comparison of Lung Cancer Screening Guidelines.

Organization	Screening Recommendation	Screening Interval	Eligibility Criteria	Countries That Have Adopted the Guideline
United States Preventive Services Task Force (USPSTF)	LDCT	Annually	Age 50–80 years, ≥20 pack-year smoking history, currently smoke or have quit within the past 15 years	United States
National Comprehensive Cancer Network (NCCN)	LDCT	Annually	Age 50–80 years, ≥20 pack-year smoking history, currently smoke or have quit within the past 15 years	United States
American Cancer Society (ACS)	LDCT	Annually	Age 50–80 years, ≥20 pack-year smoking history, currently smoke or have quit within the past 15 years	United States
European Society for Medical Oncology (ESMO)	LDCT, chest X-rays, or sputum cytology	Annually or biennially (varies by country)	Age 50–75, 20 pack-year smoking history, current smokers or those who quit within the past 10 years	Various European countries (e.g., The Netherlands, Germany, Italy)
International Association for the Study of Lung Cancer (IASLC)	LDCT	Annually or biennially (varies by country)	Age 50–80, 20 pack-year smoking history, current smokers or those who quit within the past 15 years	Various countries worldwide (e.g., Canada, Australia, Japan, South Korea)

**Table 3 healthcare-11-02085-t003:** Commonly used lung cancer risk prediction models.

Risk Prediction Model	Description	Applicability	Reference
National Lung Screening Trial (NLST)	Developed by the National Lung Screening Trial (NLST) to assess the risk of lung cancer using factors such as age, smoking history, and history of chronic obstructive pulmonary disease (COPD).	High-risk individuals, primarily current and former heavy smokers	[58]
Liverpool Lung Project	Developed by the Liverpool Lung Project, this model incorporates age, smoking history, family history, and other risk factors to estimate lung cancer risk.	General population and family history of lung cancer	[59]
PLCO Cancer Screening Trial	Developed by the Prostate, Lung, Colorectal, and Ovarian (PLCO) Cancer Screening Trial, this model predicts the risk of lung cancer based on factors such as age, smoking status, and history of COPD.	General population	[58]
I-ELCAP	Developed by the International Early Lung Cancer Action Program (I-ELCAP), this model assesses the risk of lung cancer based on factors such as age, smoking history, family history, and nodule characteristics.	High-risk individuals, primarily current and former smokers	[64]
Bach Model	Developed by Bach et al., this model estimates the risk of lung cancer based on age, smoking history, family history, and other factors.	General population	[65]
ACS Lung Cancer Risk Prediction Model	Developed by the American Cancer Society (ACS), this model estimates the risk of lung cancer based on factors such as age, smoking history, exposure to secondhand smoke, and occupational exposure to carcinogens.	General population, primarily current and former smokers	[66]

**Table 4 healthcare-11-02085-t004:** The main findings of cost-effectiveness studies.

Reference	Country	Result	Cost/Effectiveness
[68]	US	ICER USD 116,300 per QALY gained: current smokers ICER USD 558,600 per QALY gained: quitting smokers ICER USD 2,322,700 per QALY gained: former smokers	Yes
[69]	US	The ICERs were USD 52,000 per life-year gained (95% CI, 34,000 to 106,000) and USD 81,000 per QALY gained (95% CI, 52,000 to 186,000).	Yes
[70]	US	The strategy with a 1.2% risk threshold had an ICER of USD 94,659 (model range, USD 72,639 to USD 156,774)	Yes
[71]	US	Lung cancer screening programs incorporating such a hypothetical diagnostic biomarker with a medium sensitivity profile would be cost-effective if the biomarker cost was USD 250 or less, using a willingness-to-pay threshold of USD 100,000 per QALY.	Yes
[72]	US	A mean ICER of USD 72,564 (range across four models, USD 59,493–USD 85,837) per QALY gained was detected.	Yes
[73]	Australia	The ICER for lung screening compared to usual care in the NELSON-based scenario was AUD 39,250 (95% CI AUD 18,150–108,300) per quality-adjusted life-year (QALY), lower than the NLST-based estimate (ICER = USD 76,300, 95% CI USD 41,750–236,500	Yes
[74]	US	The National Lung Screening Trial (NLST), Centers for Medicare & Medicaid Services (CMS), and U.S. Preventive Services Task Force (USPSTF) screening strategies were cost-effective, with ICERs averaging USD 49,200, USD 68,600, and USD 96,700 per QALY, respectively.	Yes
[75]	Spain	In the base-case, an increase of 4.80 quality-adjusted life-years (QALY) per patient was obtained, resulting in an incremental cost-effectiveness ratio of EUR 2345/QALY.	Yes
[76]	US	Screening with LDCT increased lifetime costs by USD 1089 versus screening with chest X-ray, yielding an ICER for LDCT of USD 37,000/LY gained or USD 60,000/QALY gained in the overall NLST cohort.	Yes
[77]	UK	An incremental cost-effectiveness ratio of GBP 10,069/QALY was found.	Yes
[78]	Germany	An incremental cost-effectiveness ratio of EUR 30,291/QALY was reported.	Yes
[79]	China	ICER ranged from CNY 119,974.08 to CNY 614,167.75 per QALY gained relative to non-screening.	Yes
[80]	Japan US	LDCT yielded the greatest benefts with the lowest cost in Japan, but the ICERs of LDCT compared with CXR were USD 3,001,304 per QALY gained for American men and USD 2,097,969 per QALY gained for American women. Cost-efectiveness was sensitive to the incidence of lung cancer. LDCT was cost-effective (99.3–99.7%) for Japanese people, no screening was cost-effective (77.7%) for American men, and CXR was cost-effective (93.2%) for American women.	Yes for Japan No for US
[81]	Italy	Base-case incremental cost-effectiveness ratios were EUR 3297 and EUR 2944 per quality-adjusted life-year and life-year gained, respectively.	Yes

## Data Availability

The data are contained within the article.

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
