# Peer review of "Systematic Review of Lung Cancer Screening: Advancements and Strategies for Implementation"

_healthcare, 2023, doi:10.3390/healthcare11142085_

Round 1
Reviewer 1 Report
This is a valuable study for screening lung cancer.
There is only one concern about this article: The aim of this review was to identify key areas for consideration in establishing an organized public health screening program to enhance early diagnosis and survival rates of lung cancer in Italy. Are these “key areas” are identified solely through qualitative analysis? These “key areas” seem to be well known in lung cancer screening. Can these “key areas” be quantitatively analyzed? Or make a ranking of importance?
I have no specific comments on language.
Reviewer 2 Report
The abstract must be shortened and must not contain headings.
Introduction:
A very good introduction.
Line 52: In the year 2023 alone – the aspect of the year must be clarified given that it is not concluded. Is it the retrospect published in 2023 regarding the year 2022?
Line 108 – this study, not paper
Material and method:
Study selection criteria
Were the two reviewers (D.A. and M.F.P) the same as the two double blind re-examiners?
In some stated studies, it was observed that the screening gave results in terms of detection, but not in terms of patient mortality. Has this idea of the lack of yield in terms of mortality been analyzed? What were the explanations?
nodules or tumors that are not cancerous - even if the nodules are not cancerous, this does not mean that they should not be investigated. Rephrase this idea. It is correct regarding the subject of the study (lung cancer), but the previously mentioned aspect should not be neglected.
The discussions are very well structured.
To the conclusions, I would add a synthesis of the idea from paragraph 686-692.
The text still needs small edits (pay attention to points, capital letters and the place of references)
The references are solid and current.
Minor editing of English language required.
Reviewer 3 Report
I do not agree when the authors said that " those who are unable to undergo surgery may not 302 benefit from screening", since SABR (or SBRT) is now a valid alternative to surgery for early-stage NSCLC, with good tolerabilty for the patients and similiar outcome in terms of OS and DFS.
So patients unfit for surgery may benefit from LDCT screening programs, because they may be offered an alternative radical and non-invasive treatment such as stereotactic ablative radiotherapy (SABR).
Reviewer 4 Report
This paper aims to provide a comprehensive systematic review of lung cancer screening, including its benefits, risks, and limitations. It also discusses the current state of lung cancer screening programs worldwide, evaluates the evidence for effectiveness and cost-effectiveness, and explores different implementation strategies. A literature search yielded 61 scientific papers, highlighting the effectiveness of low-dose computed tomography (LDCT) screening in reducing mortality among high-risk populations.
Review papers can be really useful but it is not easy to write a good review paper. The title of this paper sounds interesting but its content can be improved.
1- It can be useful if a chart (graph) will be added to show "the number of papers related with this topic per year".
A search engine (such as Web of Science, Scopus, Google Scholar, Science Direct) with a query can be used to obtain the counts.
2- Some application examples can also be given.
3- Important, popular and powerfull recent techniques are missing such as Data Mining, Deep Learning, etc.
4- A single and integrated abstract can be written, instead of four separate parts:
- Background
- Methods
- Results
- Conclusions
5- A keyword related to health or healthcare can be added.
6- The possible future works related to the subject can be written.
7- The organization of the paper (the structure of the manuscript) should be written at the end of the "Introduction" section.
For example: "Section 2 presents ... Section 3 gives ...."
